# Ruthenium, Not Carbon Monoxide, Inhibits the Procoagulant Activity of *Atheris*, *Echis*, and *Pseudonaja* Venoms

**DOI:** 10.3390/ijms21082970

**Published:** 2020-04-23

**Authors:** Vance G. Nielsen

**Affiliations:** Department of Anesthesiology, University of Arizona College of Medicine, Tucson, AZ 85719, USA; vgnielsen333@gmail.com

**Keywords:** hemotoxin, procoagulant, ruthenium, carbon monoxide releasing molecule, thrombelastography

## Abstract

The demonstration that carbon monoxide releasing molecules (CORMs) affect experimental systems by the release of carbon monoxide, and not via the interaction of the inactivated CORM, has been an accepted paradigm for decades. However, it has recently been documented that a radical intermediate formed during carbon monoxide release from ruthenium (Ru)-based CORM (CORM-2) interacts with histidine and can inactivate bee phospholipase A_2_ activity. Using a thrombelastographic based paradigm to assess procoagulant activity in human plasma, this study tested the hypothesis that a Ru-based radical and not carbon monoxide was responsible for CORM-2 mediated inhibition of *Atheris,*
*Echis*, and *Pseudonaja* species snake venoms. Assessment of the inhibitory effects of ruthenium chloride (RuCl_3_) on snake venom activity was also determined. CORM-2 mediated inhibition of the three venoms was found to be independent of carbon monoxide release, as the presence of histidine-rich albumin abrogated CORM-2 inhibition. Exposure to RuCl_3_ had little effect on *Atheris* venom activity, but *Echis* and *Pseudonaja* venom had procoagulant activity significantly reduced. In conclusion, a Ru-based radical and ion inhibited procoagulant snake venoms, not carbon monoxide. These data continue to add to our mechanistic understanding of how Ru-based molecules can modulate hemotoxic venoms, and these results can serve as a rationale to focus on perhaps other, complementary compounds containing Ru as antivenom agents in vitro and, ultimately, in vivo.

## 1. Introduction

The use of carbon monoxide releasing molecules (CORMs) to deliver carbon monoxide (CO) in a site-directed fashion to presumably alter heme-modulated systems has been part of experimental designs for decades, with hundreds of manuscripts incorporating this methodology. The key element of the paradigm that implicates CO as the mechanism behind the effects of CORMs is the determination that the inactivated releasing molecule (iRM), the portion of the CORM that remains after CO release, has no effect or a different effect on the system tested with the CORM compared to the anticipated CO effect. This laboratory has used this CORM-based paradigm for the past few years to demonstrate that CO inhibited the various procoagulant and anticoagulant activities of hemotoxic venoms and enzymes collected from dozens of snake and lizard species [1,2,3,4,5,6,7,8,9,10,11,12,13]. The particular CORM used was CORM-2 (tricarbonyldichlororuthenium (II) dimer) [1,2,3,4,5,6,7,8,9,10,11,12,13]. The presumed mechanism was that CO must be interacting with a cryptic heme group attached to the various venoms and enzymes or in some other way interacting with these diverse enzymes and venoms [1,2,3,4,5,6,7,8,9,10,11,12,13]. Thus, this potentially heme-based, CO-modulated paradigm of snake venom activity seemed plausible with the aforementioned paradigm of CORM-CO release-inert iRM interactions with target molecules.

However, cracks in the edifice of this paradigm began to appear in the year 2017 with the publication of a work that demonstrated CO-independent inhibition of K^+^ channels with a putative Ru-based radical formed from CORM-2 during CO release and likely prior to formation of its iRM [14]. To substantiate this claim, the authors demonstrated that free histidine or albumin, which is resplendent with histidine residues, quenched the inhibition of potassium channels by CORM-2. Furthermore, using mass spectroscopy, the authors demonstrated histidine-Ru-based radical formation following exposure of free histidine with CORM-2 [14]. Lastly, using other various CORMs with other metal centers, the authors demonstrated no CO effects on the channel assessed [14]. This laboratory became aware of this work recently, and using a similar approach, demonstrated an identical outcome wherein the anticoagulant activity of the purified phospholipase A_2_ (PLA_2_) of *Apis mellifera* venom was inhibited by CORM-2 in a CO-independent, albumin-inhibitable fashion [15]. Furthermore, we recently demonstrated that the anticoagulant metalloproteinases of mamba venoms are inhibited by CORM-2 in a CO-independent, albumin-inhibitable manner [16]. Taken as a whole, it was entirely possible that Ru-based interactions with venom proteins could be responsible for the inhibition noted in our previous works [13]; and critically, if Ru-based modifications were the underpinning of such inhibition rather than the interaction of CO with a heme group, then Ru-based CORMs could well serve as permeant antivenom agents. The importance of these line of investigation involving ion channels [14], phospholipase A_2_ [15], and metalloproteinases [16] is that they lay the foundation to seriously reconsider the paradigm that Ru-based CORMs affect systems as simple as enzymes to as complex as whole animal models of disease in CO-independent ways—potentially affecting the interpretation of data contained in several hundred manuscripts.

While it is unreasonable to reassess all previous venoms inhibited by CORM-2 to determine if a Ru-based radical rather than CO was mediating the inhibition [1,2,3,4,5,6,7,8,9,10,11,12,13], assessing a few representative venoms would be of benefit. To this end, three procoagulant venoms derived from diverse species from Africa and Australia were selected that have already been characterized as inhibited by CORM-2 but not by its iRM by this laboratory [8,10]. The species chosen are displayed in Table 1, and the venom proteomes of these particular and snakes within the same genus are similar in terms of presence of snake venom serine proteases (SVSP), snake venom metalloproteinases (SVMP), and PLA_2_ [17,18,19,20,21]. Fortuitously, archived aliquots of these three venoms that were never thawed or used in the original studies [8,10] were maintained at −80 °C and were available for the present investigation to test the hypothesis that inhibition by ruthenium molecular species and not carbon monoxide may be the mechanism by which these procoagulant venoms were inhibited by CORM-2.

Considering the aforementioned, the present investigation had the following goals. First, determination of inhibition of the procoagulant activities of these venoms by their exposure in isolation to CORM-2 in the absence or presence of albumin was to be performed as previously described with bee venom PLA_2_ [15] and mamba venom [16]. Second, to further assess if Ru-based molecules may affect venom procoagulant activity, the three venoms were exposed to equimolar concentrations of ruthenium chloride (RuCl_3_) which contains a Ru^+3^ state compared to the Ru^+2^ state of CORM-2. Compounds incorporating Ru^+3^ more complex than RuCl_3_ have been demonstrated to covalently bond to histidine residues in several proteins [22,23], thus offering the possibility that RuCl_3_ could interact with histidine-bearing venom enzymes. Critically, the use of RuCl_3_ in this investigation was purely to provide mechanistic insight, and as there is no clinical indication to administer it to humans or any other species, I am not advocating it as a therapeutic option. As previously described [1,2,3,4,5,6,7,8,9,10,11,12,13,15,16], changes in procoagulant activity were assessed with human plasma via changes in coagulation kinetics determined with thrombelastography.

## 2. Results

### 2.1. Assessment of the CO-Independent, Ru-Dependent Inhibition of CORM-2 on Procoagulant Activity of *A. nitschei*, *E. leucogaster*, and *P. textilis* Venoms Assessed with Thrombelastography

The subsequent results were obtained using concentrations of the aforementioned venoms previously published [8,10]; specifically, *A. nitschei* and *E. leucogaster* venoms had a final concentration of 1 µg/mL in the plasma mixtures whereas *P. textilis* venom was at a final concentration of 100 ng/mL. Venom concentrations were originally chosen based on a performance basis wherein the activation of coagulation by the venom statistically exceeded the activation observed by contact activation with thrombelastographic cup and pin contact with plasma as previously described [8,10]. All venom solutions without or with chemical additions in isolation were added as a 1% addition to the plasma mix used in our thrombelastographic system [1,2,3,4,5,6,7,8,9,10,11,12,13]. This dilution is critical, as it reduces the concentration of CORM-2 to 1 µM, a concentration at which this compound does not affect coagulation kinetics [9]. The thrombelastographic model describes coagulation kinetics with the following three variables: time to maximum thrombus generation (TMRTG, minutes—a measure of time to onset of coagulation), maximum rate of thrombus generation (MRTG, dynes/cm^2^/s—a measure of the velocity of clot growth) and total thrombus generation (TTG, dynes/cm^2^—a measure of clot strength). The results of exposing the three venoms to CORM-2 in the absence or presence of 5% human albumin (*n* = 6 for all conditions) are depicted in Figure 1, Figure 2 and Figure 3.

All three venoms behaved kinetically as procoagulants as previously noted [8,10], significantly decreasing TMRTG and increasing MRTG values. Similarly, exposure of the venoms to CORM-2 in PBS significantly attenuated procoagulant activity [8,10]. However, when the venoms were exposed to CORM-2 dissolved in 5% human albumin, procoagulant activity was uninhibited. These results strongly support a CO-independent, Ru-dependent mechanism of inhibition of venom procoagulant activity by CORM-2. Results concerning the effects of RuCl_3_ on plasmatic coagulation and venom procoagulant activity are subsequently presented.

### 2.2. Assessment of the Effects of RuCl_3_ on Human Plasmatic Coagulation—Roles of Concentration and Vehicle

Prior to experimentation with venom, preliminary investigation demonstrated that the expected residual 1 µM concentration of RuCl_3_ dissolved in PBS increased MRTG values compared to control conditions. It has already been demonstrated that up to 10 µM CORM-2 in PBS has no effect on plasmatic coagulation kinetics [9], so a more formal determination of why RuCl_3_ displayed procoagulant properties was indicated. Given that the only compounds present with anions different from RuCl_3_ potentially available to displace Cl in the PBS used was KH_2_PO_4_ (1.5 mM) and Na_2_HPO_4_ (8.1 mM), a comparison of the effects of 1 and 10 µM RuCl_3_ (final concentration) dissolved in dH_2_O or PBS was performed in plasma as a 1% addition (*v*/*v*) with coagulation assessed via thrombelastography. The results of these experiments are displayed in Figure 4 (*n* = 6 per condition). As can be readily discerned, there appears to be a Ru-dependent, vehicle-independent significant decrease in TMRTG and increase in MRTG values when considering the two-way analysis of variance (ANOVA) results and post hoc comparison of the two concentrations of RuCl_3_ dissolved in dH_2_O. Similarly, increased RuCl_3_ concentrations significantly decrease TMRTG and increase MRTG values when PBS is the vehicle. However, and critically, the coagulation kinetic differences caused by RuCl_3_ are significantly enhanced by the fluid it is dissolved in as indicated by the two-way ANOVA significance values in each panel of Figure 4. Of interest, while TMRTG and MRTG values change in a manner indicative of procoagulation, TTG is decreased by interactions of RuCl_3_ and fluid. This thrombelastographic pattern is indicative of enhanced thrombin–fibrinogen interactions without enhanced activation of factor XIII (FXIII) [24,25].

As all of the venoms investigated over the past few years have been suspended in PBS for the purposes of preserving enzymatic function within a physiological pH, storage, and experimentation [1,2,3,4,5,6,7,8,9,10,11,12,13,15,16], it seemed prudent to continue with the use of PBS in the subsequently described experimental series assessing the effects of RuCl_3_ on the three venoms of the present work. The caveat that should be kept in mind was that small enhancements of MRTG in human plasma could be secondary to a RuCl_3_/PBS interaction when determining if RuCl_3_ inhibited venom procoagulant activity.

### 2.3. Assessment of RuCl_3_-Dependent Modulation of CORM-2 on the Procoagulant Activity of *A. nitschei*, *E. leucogaster*, and *P. textilis* Venoms Assessed with Thrombelastography

Utilizing the same general experimental approach and specific concentrations of the three venoms tested, additional aliquots of each venom was exposed to 100 µM concentrations of RuCl_3_ in PBS for 5 min prior to being placed into the plasma mixture as a 1% addition (*v*/*v*). The rationale for this concentration of RuCl_3_ was that it would be similar to that of the CORM-2 exposure experiments described in Section 2.1. The results of these experiments are displayed in Figure 5, Figure 6 and Figure 7.

Unlike in the series with CORM-2 wherein the pattern of inhibition of procoagulation was very similar among the three venoms tested, there was remarkable diversity in modulation of procoagulant activity when the venoms were exposed to RuCl_3_. *A. nitschei* venom had a very diminutive increase in TMRTG which is indicative of decreased procoagulant activity in response to RuCl_3_ exposure, but a significant increase in both MRTG and TTG is consistent with an enhancement of procoagulation following RuCl_3_ exposure. In contrast, *E. leucogaster* venom demonstrated a significant increase in TMRTG, decrease in MRTG and decrease in TTG values following RuCl_3_ exposure. Lastly, *P. textilis* venom demonstrated significant loss of procoagulant activity after RuCl_3_ exposure to a qualitatively greater extent than the other two venoms. When compared to the relatively consistent pattern and degree of procoagulant activity among the three venoms provided by CORM-2 via a presumed CO-independent/Ru-dependent mechanism, modulation of the venoms by RuCl_3_ resulted in diverse changes in venom activity.

## 3. Discussion

This investigation accomplished its objectives. The first series of experiments demonstrated that the procoagulant activity of *A. nitschei, E. leucogaster*, and *P. textilis* venom was not inhibited by CO but instead by a presumed Ru-based CORM-2 radical that likely binds to venom enzyme histidine residues, evidenced by the loss of CORM-2 mediated inhibition in the presence of histidine-rich human albumin. These studies provided excess histidines to bind with reactive Ru species formed during CO release from CORM-2 by using 5% albumin (752 µM) as the solution for isolated exposures of venom to 100 µM CORM-2. Given that CORM-2 forms 70 µM of reactive Ru species during CO release from 100 µM CORM-2 [26], and that albumin has 16 histidine residues [27], in the first series of experiments there was a 160:1 molar excess of histidine to react with Ru-based species. Thus, as was recently demonstrated with the same experimental approach with PLA_2_ derived from *Apis mellifera* venom [15] and SVMP contained within mamba venom [16], it appears that procoagulant enzymes derived from the three venoms tested are vulnerable to inhibition by a Ru-based species formed from CORM-2.

The second series of experiments provided further evidence that Ru-based molecules could modulate procoagulant snake venom enzymes, but with more variability of response to exposure to RuCl_3_ compared to the CORM-2 experiments. There were three very different degrees in increase in TMRTG values in response to RuCl_3_ exposure as seen in Figure 5. In contrast to TMRTG, in the case of MRTG values it appeared that RuCl_3_ exposure enhanced *A. nitschei* venom procoagulant activity, whereas in the cases of *E. leucogaster* and *P. textilis* venoms there was inhibition of procoagulant activity by RuCl_3_ as displayed in Figure 6. Again, as with MRTG, changes in TTG values generated by the three venoms to RuCl_3_ followed the same species-specific pattern of enhanced procoagulant activity by *A. nitschei* venom and inhibited activity of *E. leucogaster* and *P. textilis* venoms as noted in Figure 7. Given that RuCl_3_ in PBS does enhance MRTG to a small but significant extent, the MRTG results obtained with *A. nitschei* venom may be indicative of venom-independent enhancement of coagulation; however, the mixed finding of small increases in TMRTG values and increase in TTG values are likely secondary to direct modulation of *A. nitschei* venom as there were no venom-independent effects on TMRTG and TTG by RuCl_3_ (Figure 4) consistent with these venom-mediated changes. The mechanisms responsible for differential effects on the three venoms by CORM-2 (Ru^+2^) compared to RuCl_3_ (Ru^+3^) remain to be defined, but when considered as a whole, the data of the present work strongly support the concept that Ru-based molecules, and not CO, are likely responsible for the CORM-2 mediated inhibition of diverse snake and lizard venoms [1,2,3,4,5,6,7,8,9,10,11,12,13].

An unexpected finding was the procoagulant effects of RuCl_3_ on human plasma that was enhanced by having the compound dissolved in PBS as seen in Figure 4. This laboratory, using thrombelastographic methods, has already documented the effects of Fe^+3^ [28] and Cu^+2^ [29] as procoagulant and anticoagulant metals, respectively, via modulation of fibrinogen. However, neither Hg^+2^ nor Pb^+2^ at lethal concentrations were found to affect human plasmatic coagulation with the same thrombelastographic methods [30]. Thus, the discovery that a Ru^+3^ compound (RuCl_3_) at concentrations of a 1 µM affects plasmatic coagulation while a Ru^+2^ compound (inactivated CORM-2 [31,32]) needs to be at 100 µM concentrations to increase MRTG was unanticipated and interesting. The precise mechanism responsible for RuCl_3_ and its PBS induced ionic species was not defined by this investigation as it is beyond its scope.

The utilization of thrombelastography to ask and answer molecular biology questions has been occurring for well over two decades, with numerous articles addressing hematological matters in several fields of investigation. It is critical to note that it is not the technique in the mechanical sense, but rather the parameters assessed and the composition of the sample analyzed that transform descriptive data that is phenomenological to parametric data that provides mechanistic insight into a focused experimental approach to testing molecular biological hypotheses. For example, the venoms assessed in this work are thrombin-generators that either activate prothrombin directly or indirectly by activating immediate precursor serine proteases in human plasmatic coagulation pathways. This feature of the venoms is best tested in a system with a relatively weak endogenous thrombin-generating scenario such as that associated with contact protein activation via interaction with the plastic surfaces of the thrombelastographic cup and pin. This allows the venom to outcompete such comparatively weak contact protein activation, permitting one the ability to assess the procoagulant activity in the presence or absence of prior isolated exposure to inhibitors or other relevant modulators. Use of the parameters TMRTG, MRTG, and TTG permits the use of parametric statistics as these expressions of clot initiation, velocity of growth, and final strength are not relatively qualitative as the unprocessed thrombelastographic tracing or nonparametric parameters such as the angle (°) or maximum amplitude (mm) [33]. Furthermore, the use of plasma, and not whole blood with intact platelet activity, simplifies the output of the experiment wherein the coagulation kinetics are dependent on the invariant fibrinogen concentration and FXIII activity that are critical as previously published [24,25]. When summated, this experimental system will provide unambiguous data that is highly reproducible. The introduction of the variation of platelet concentration, variability in platelet glycoprotein IIb/IIIa receptor content, red blood cell concentration, artificially created blood flow models, etc., provide no additional mechanistic insight and instead introduce multiple confounding effects that may preclude testing the hypothesis of the present work. Similarly, utilizing standard coagulation assessments such as activated partial thromboplastin time (contact protein activation) or prothrombin time (tissue factor activated) simply introduce increased thrombin generation via activation of plasmatic contact protein and factor VII pathways, respectively—which would only compete with the venoms tested and provide no mechanistic insight. As was mentioned previously in this passage, the goal was to create an environment wherein the thrombin-generating activity of the venoms would not be confounded by any other coagulation activation. Taken as a whole, the experimental approach taken by the present investigation was designed to produce the unambiguous data presented to vigorously and conclusively test the hypothesis espoused.

In conclusion, the present work determined that CORM-2 inhibited three already characterized procoagulant venoms via a CO-independent, likely Ru-based radical-dependent mechanism. Furthermore, a Ru^+3^-based ion also differentially affected the procoagulant activity of the venoms tested. With regard to utilization of Ru-based compounds as antivenoms, while the inhibitory effects on venom hemotoxic activity was inhibited by albumin in vitro at concentrations observed in vivo in the circulation, it is planned to administer such Ru-based compounds at concentrations at least 10-fold greater at the bite wound. Venom exposure to such concentrations has already been performed with neutralization in vivo in rabbits as recently noted [34]. Furthermore, these results continue to broaden the questioning of the effects of CORM-2 being CO-based, supporting the concept that the several hundred investigations conducted over the past few decades may include situations wherein Ru-based radical activity may be responsible for the effects of CORM-2. Lastly, these data continue to add to our mechanistic understanding of how Ru-based molecules can modulate hemotoxic venoms, and these results can serve as a rationale to perhaps focus on other complementary compounds containing Ru as antivenom agents in vitro and ultimately in vivo.

## 4. Materials and Methods

### 4.1. Chemicals and Human Plasma

Lyophilized *A. nitschei*, *E. leucogaster*, and *P. textilis* venoms were originally obtained from Mtoxins (Oshkosh, WI, USA). Venoms were dissolved into calcium-free phosphate buffered saline (PBS, Millipore Sigma, Saint Louis, MO, USA) to a final 50 mg/mL concentration, aliquoted, and maintained at −80 °C. The aliquots used came from the same lot published previously [8,10]. Dimethyl sulfoxide (DMSO), tricarbonyldichlororuthenium (II) dimer (CORM-2), and RuCl_3_ were obtained from Millipore Sigma (Saint Louis, MO, USA). Human albumin solution (5% in 0.9% NaCl) was obtained from Grisfols Biologicals Inc. (Los Angeles, CA, USA). Calcium chloride (200 mM) was obtained from Haemonetics Inc. (Braintree, MA, USA). Pooled normal human plasma that was sodium citrate anticoagulated and maintained at −80 °C was obtained from George King Bio-Medical (Overland Park, KS, USA). This plasma is a commercial product collected from consented, anonymous, and compensated healthy donors by the vendor, so no further consent is needed to be obtained by end users.

### 4.2. Thrombelastographic Analyses

The volumes of subsequently described plasmatic and other additives summed to a final volume of 360 µL. Samples were composed of 320 µL of plasma; 16.4 µL of PBS, 20 µL of 200 mM CaCl_2_, and 3.6 µL of PBS, RuCl_3_, or venom solution mixture, which were pipetted into a disposable cup in a thrombelastograph^®^ hemostasis system (Model 5000, Haemonetics Inc., Braintree, MA, USA) at 37 °C, and then rapidly mixed by moving the cup up against and then away from the plastic pin five times. The PBS, RuCl_3_, or venom solution mixtures was always the last constituent added prior to mixing and data collection. The following viscoelastic parameters described previously [1,2,3,4,5,6,7,8,9,10,11,12,13] were measured: time to maximum rate of thrombus generation (TMRTG): this is the time interval (minutes) observed prior to maximum speed of clot growth; maximum rate of thrombus generation (MRTG): this is the maximum velocity of clot growth observed (dynes/cm^2^/s); and total thrombus generation (TTG, dynes/cm^2^), the final viscoelastic resistance observed after clot formation. Data were collected until the clot strength reached its final plateau (maximum amplitude) that was stable for 3 min.

The concentrations of venom that were used were as previously indicated in results and past manuscripts [8,10].

### 4.3. CORM-2 Addition Experiments

In experiments with CORM-2 the conditions utilized were: (1) control condition—no venom, DMSO 1% addition (*v*/*v*) in PBS; (2) V condition—venom at the concentration determined in preliminary studies, DMSO 1% addition (*v*/*v*) in PBS; (3) VC condition—venom at the concentration as in condition 2, CORM-2 1% addition in DMSO in PBS (100 µM); (4) VC+A condition—venom and CORM-2 1% addition in DMSO in 5% human albumin (100 µM final concentration). Solutions were incubated for 5 min at room temperature, and then 3.6 µL of one of these solutions was added to the plasma sample in the plastic cup.

### 4.4. RuCl_3_ Addition Experiments

In preliminary experiments with RuCl_3_, it was determined that the final concentration of 1 µM increased MRTG. Thus, experiments wherein RuCl_3_ was dissolved in either dH_2_O or PBS were conducted with the final concentration of RuCl_3_ in plasma being 1 or 10 µM. Data were collected until maximum amplitude was observed.

In venom exposure experiments, aliquots of venom dissolved in PBS were exposed to 100 µM RuCl_3_ previously dissolved in PBS for 5 min at room temperature prior to being placed in the plasma mixture in the thrombelastographic cup as a 1% (*v*/*v*) addition. Data were collected until maximum amplitude was observed.

### 4.5. Graphics and Statistical Analyses

Data are presented as mean ± SD. Graphics were generated with a commercially available program (Origen2020, OrigenLab Corporation, Northampton, MA, USA). Experimental conditions were composed of *n* = 6 replicates per condition as this provides a statistical power >0.8 with *p* < 0.05 utilizing these techniques [1,2,3,4,5,6,7,8,9,10,11,12,13]. A statistical program was used for one-way analyses of variance (ANOVA) comparisons between conditions, followed by Holm–Sidak post hoc analysis (SigmaPlot 14, Systat Software, Inc., San Jose, CA, USA). *p* < 0.05 was considered significant.

## Figures and Tables

**Figure 1 ijms-21-02970-f001:**
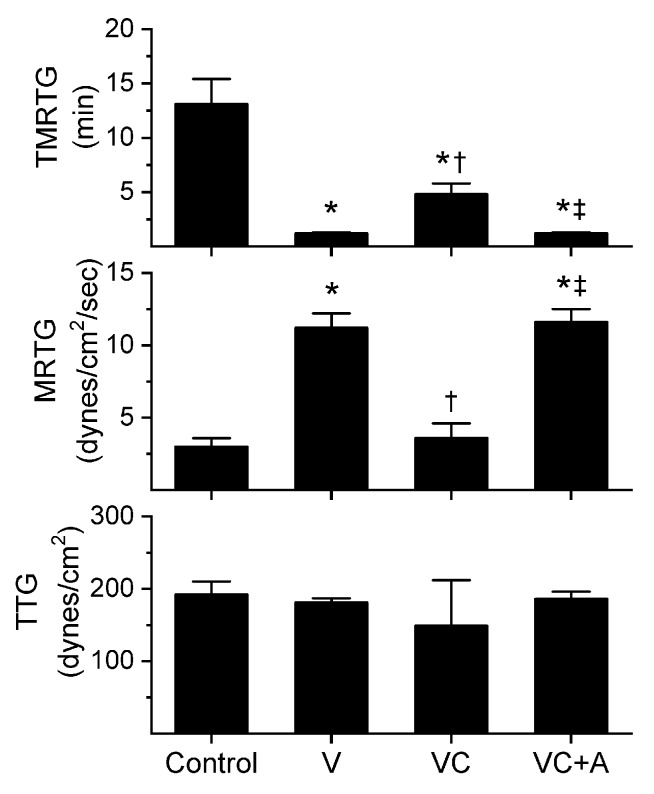
Procoagulant activity of *A. nitschei* venom in plasma after exposure to CORM-2 without or with albumin in isolation. Data is presented as mean ± SD. Control = no additives; V = venom; VC = V with 100 µM CORM-2 in PBS; VC + A = V with 100 µM CORM-2 in albumin. * *p* < 0.05 vs. control; † *p* < 0.05 vs. V; ‡ *p* < 0.05 vs. VC via one-way analysis of variance (ANOVA) with Holm–Sidak post hoc test.

**Figure 2 ijms-21-02970-f002:**
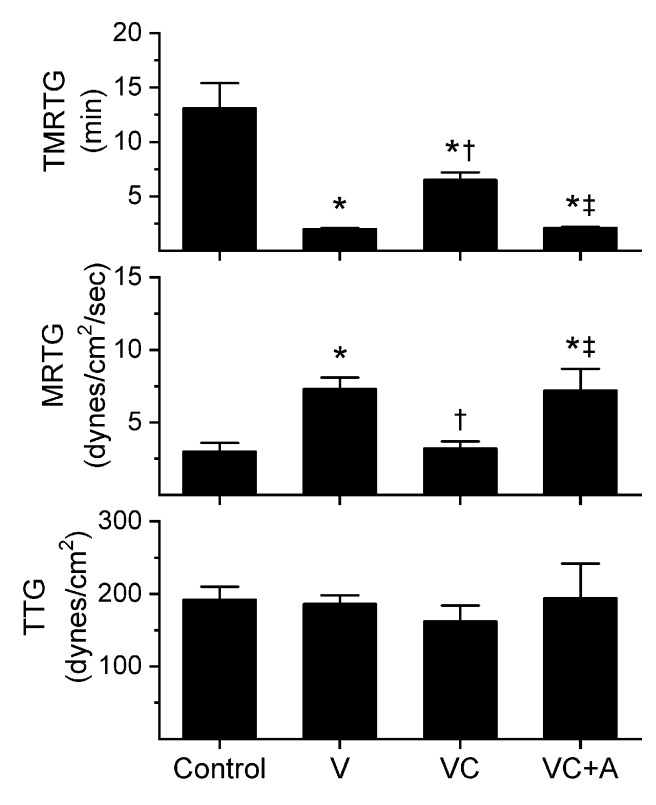
Procoagulant activity of *E. leucogaster* venom in plasma after exposure to CORM-2 without or with albumin in isolation. Data are presented as mean ± SD. Control = no additives; V = venom; VC = V with 100 µM CORM-2 in PBS; VC + A = V with 100 µM CORM-2 in albumin. * *p* < 0.05 vs. control; † *p* < 0.05 vs. V; ‡ *p* < 0.05 vs. VC via one-way ANOVA with Holm–Sidak post hoc test.

**Figure 3 ijms-21-02970-f003:**
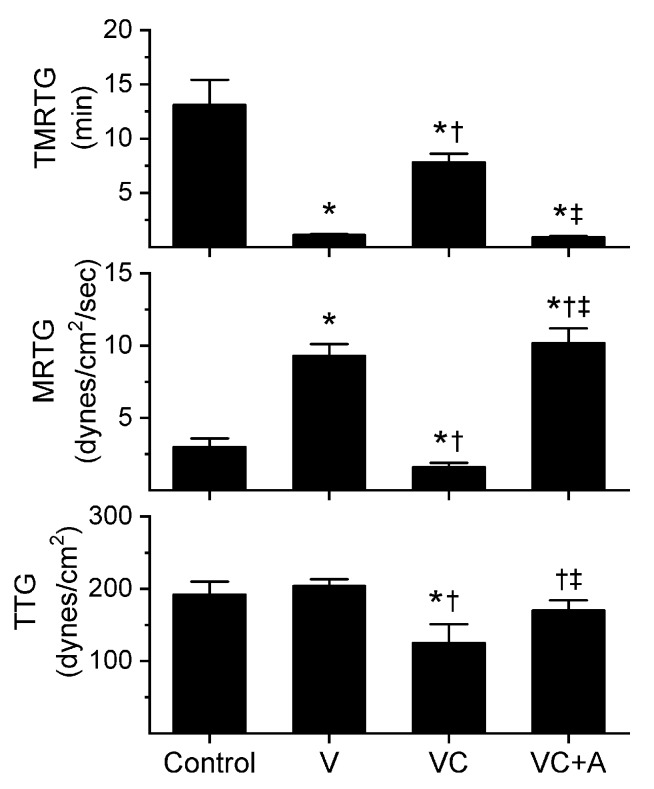
Procoagulant activity of *P. textilis* venom in plasma after exposure to CORM-2 without or with albumin in isolation. Data are presented as mean ± SD. Control = no additives; V = venom; VC = V with 100 µM CORM-2 in PBS; VC + A = V with 100 µM CORM-2 in albumin. * *p* < 0.05 vs. control; † *p* < 0.05 vs. V; ‡ *p* < 0.05 vs. VC via one-way ANOVA with Holm–Sidak post hoc test.

**Figure 4 ijms-21-02970-f004:**
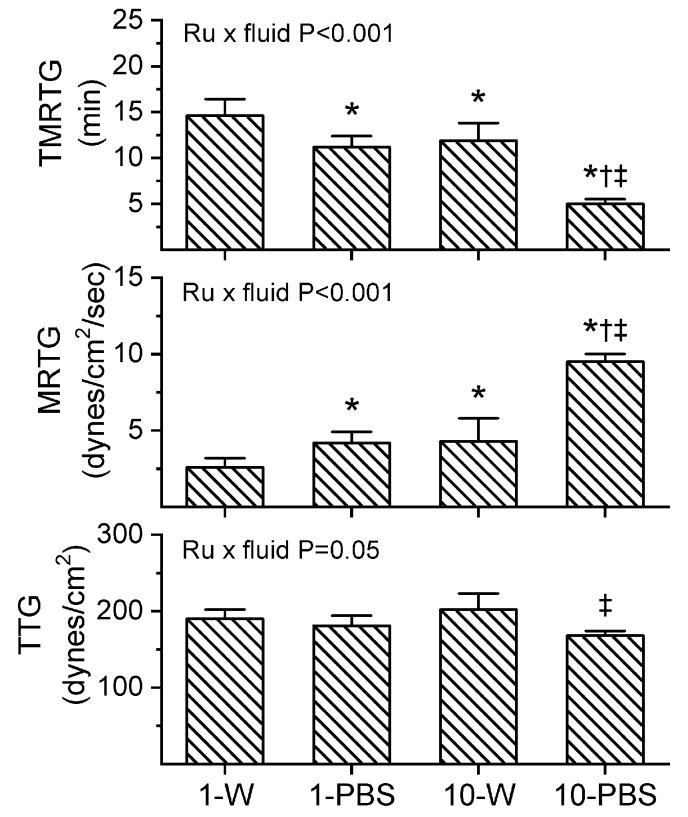
Interactions of RuCl_3_ concentration and fluid within which it is dissolved. Data are mean ± SD. 1-W = 1 µM RuCl_3_ in dH_2_O; 1-PBS = 1 µM RuCl_3_ in PBS; 10-W = 10 µM RuCl_3_ in dH_2_O; 10-PBS = 10 µM RuCl_3_ in PBS. * *p* < 0.05 vs. 1-W; † *p* < 0.05 vs. 1-PBS; ‡ *p* < 0.05 vs. 10-W via two-way ANOVA with Holm–Sidak post hoc test. Two-way ANOVA results for interaction of RuCl_3_ concentration and fluid are indicated within each panel.

**Figure 5 ijms-21-02970-f005:**
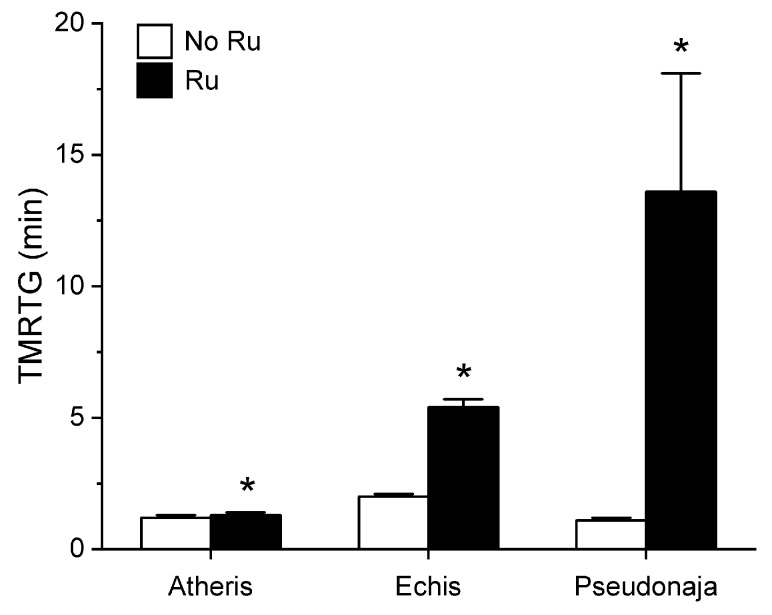
Effect of exposure of *A. nitschei, E. leucogaster,* and *P. textilis* venom to 100 µM RuCl_3_ in PBS on TMRTG values in human plasma. Data are presented as mean ± SD. White bars = no RuCl_3_ exposure; black bars = 100 µM RuCl_3_ in PBS exposure. * *p* < 0.05 vs. No RuCl_3_ in PBS exposure via two-tailed, unpaired *t*-test.

**Figure 6 ijms-21-02970-f006:**
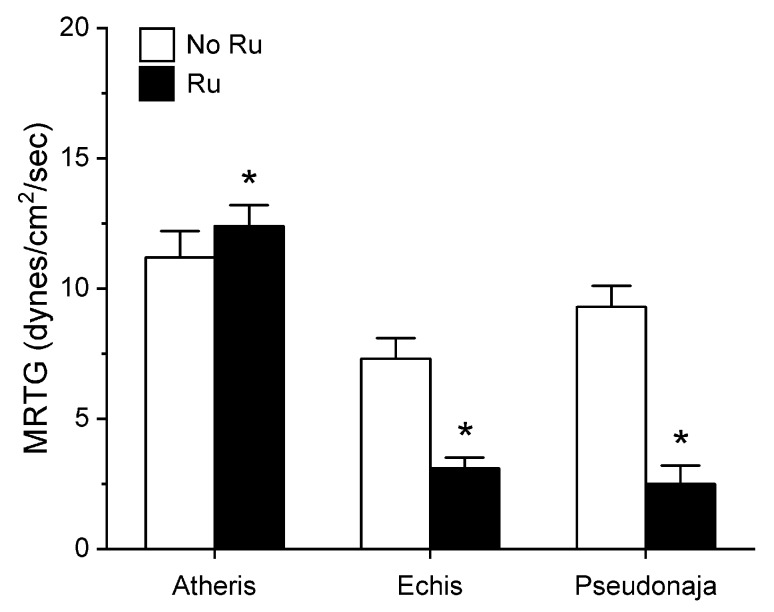
Effect of exposure of *A. nitschei, E. leucogaster,* and *P. textilis* venom to 100 µM RuCl_3_ in PBS on MRTG values in human plasma. Data are presented as mean ± SD. White bars = no RuCl_3_ exposure; black bars = 100 µM RuCl_3_ in PBS exposure. * *p* < 0.05 vs. no RuCl_3_ in PBS exposure via two-tailed, unpaired *t*-test.

**Figure 7 ijms-21-02970-f007:**
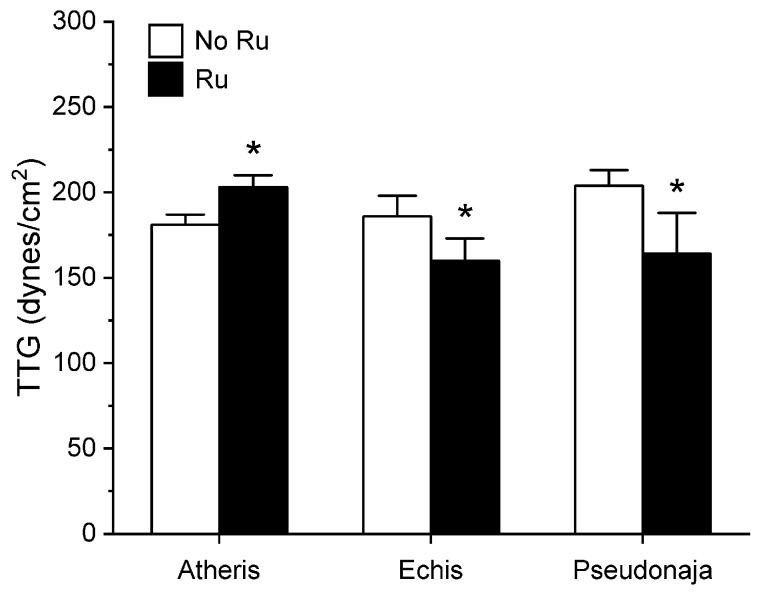
Effect of exposure of *A. nitschei, E. leucogaster,* and *P. textilis* venom to 100 µM RuCl_3_ in PBS on MRTG values in human plasma. Data are presented as mean ± SD. White bars = no RuCl_3_ exposure; black bars = 100 µM RuCl_3_ in PBS exposure. * *p* < 0.05 vs. no RuCl_3_ in PBS exposure via two-tailed, unpaired *t*-test.

**Table 1 ijms-21-02970-t001:** Properties of procoagulant snake venoms investigated.

Species	Common Name	Proteome	CORM-2/iRM Inhibition
*Atheris nitschei* [16]	Great Lakes Bush Viper	SVSP, SVMP, PLA_2_	Yes/No
*Echis leucogaster* [17,18,19]	White-Bellied Carpet Viper	SVSP, SVMP, PLA_2_	Yes/No
*Pseudonaja textilis* [20]	Eastern Brown Snake	SVSP, SVMP, PLA_2_	Yes/No

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
