# Peer review of "Ruthenium, Not Carbon Monoxide, Inhibits the Procoagulant Activity of Atheris, Echis, and Pseudonaja Venoms"

_ijms, 2020, doi:10.3390/ijms21082970_

Round 1

Reviewer 1 Report

This is interesting study, but only for the limited number of readers. Nevertheless it might stimulate further studies of Ru-based antivenom agents.

Was CORM-2 effect on thromboelastography excluded?

Author Response

“This is interesting study, but only for the limited number of readers. Nevertheless it might stimulate further studies of Ru-based antivenom agents.” 

I thank the reviewer for the kind comment.

“Was CORM-2 effect on thromboelastography excluded?” 

It was.  The 1% addition of venom mixture reduced CORM-2 to concentrations already documented to have no affect on coagulation.  This is now noted at the beginning of Results with reference 9 cited.

Reviewer 2 Report

Review of the manuscript: Ruthenium, not carbon monoxide, inhibits the 2 procoagulant activity of Atheris, Echis and Pseudonaja venoms

The manuscript describes the effects of Ru-based radical on CORM-2 mediated inhibition of Atheris, Echis and Pseudonaja species snake venoms.

Although the topic of the paper appears to be interesting, the author did several mistakes which are listed below. Unfortunately the study lacks advanced methods, and relies only on simple methodology.

  1. The author should explain the rationale for undertaking the study. Please state why do you search for inhibition of anticoagulant activity of snake venom?
  2. Please state also what the future plans are regarding these studies. What the author plans to achieve?
  3. The author should also perform basic coagulological testes, like PT or APTT.
  4. The studies should be performed in more advanced systems, e.g. in blood flow condition to see how plasma and platelet haemostasis is affected.
  5. Minor point: please remove double spaces through the whole manuscript.

Author Response

“The manuscript describes the effects of Ru-based radical on CORM-2 mediated inhibition of Atheris, Echis and Pseudonaja species snake venoms.”

“Although the topic of the paper appears to be interesting, the author did several mistakes which are listed below. Unfortunately the study lacks advanced methods, and relies only on simple methodology.”

I would like to convince the reviewer that the methods used were sophisticated based on the manner that the venoms were exposed to conditions wherein different ruthenium species could interact with them and profoundly affect their procoagulant activity. By analogy, a spectrophotometer is not particularly remarkable as an instrument; however, if the samples placed in it have been manipulated to provide an unambiguous mechanistic answer to a molecular biology question, then the method – not the instrument – is advanced. This is precisely what was performed in the present investigation.

While not explicitly stated by the reviewer, it appears that methods utilizing thrombelastography are being criticized as not being “advanced”. I have developed several thrombelastographic assays and methodological approaches over the past 20 years, and have published 117 manuscripts concerning relevant coagulation and fibrinolytic matters with them in both basic science and clinical studies.  Two of these manuscripts using thrombelastographic methods were published in this journal within the last seven months.  The methods used in the present study are among the most sophisticated involving the thrombelastograph, and I would encourage the reviewer to reexamine my manuscript and some of the relevant citations. In sum, there are no other methodological approaches that allow clear observations to be made of either procoagulant or anticoagulant venoms by documenting coagulation kinetics in the manner thrombelastography can by changes in clot initiation, propagation and final strength.

“The author should explain the rationale for undertaking the study. Please state why do you search for inhibition of anticoagulant activity of snake venom?”

I would appeal to the reviewer to read the Introduction of my paper.  The rationale for the study is clearly outlined in the four paragraphs of this section, with the goals of the study explicitly stated.  My investigation was designed to determine if ruthenium, not carbon monoxide, was responsible for the inhibition of the dozens of hemostatic venoms studied in my lab over the past three years.  All of the methods, data, and relevant references are provided by my manuscript.  Lastly, as the manuscript title and body of the work explicitly states, I was studying procoagulant – not anticoagulant – venom.

“Please state also what the future plans are regarding these studies. What the author plans to achieve?”

I again would entreat to read the Conclusion section of my manuscript.  The answer to these questions is clearly stated, with a summary paragraph at the end of the section indicating future directions.  My data point to exploring ruthenium species as widely effective antivenoms, and this new direction will be exciting to toxinologists.

“The author should also perform basic coagulological testes, like PT or APTT.”

I must respectfully disagree with the reviewer. The venoms tested are basically prothrombin activating, which would trump activation of the FVII pathway with tissue factor or the FXII pathway with celite or kaolin. The use of either PT or aPTT would provide no insight into the questions asked by this investigation, and in the presence of the venoms, the values of PT or aPTT would be abnormally decreased. Thrombelastography wherein the thrombin generation is via weak FXII activation in the recalcified plasma by plastic pin and cup contact is an excellent milieu to assess changes in vigorous prothrombin activating venom activities before and after inhibition by ruthenium species.

“The studies should be performed in more advanced systems, e.g. in blood flow condition to see how plasma and platelet haemostasis is affected.”

I again respectfully disagree with the reviewer.  The introduction of a blood flow model would provide no meaningful insight into the molecular interactions of the venoms tested with ruthenium species – there would more likely be confounding variables introduced that would prevent the clear changes in venom activity caused by ruthenium species as determined by thrombelastography.  Further, the introduction of whole blood and platelets would also obscure such interactions compared to the utilization of an already very well characterized plasma-based system with the thrombelastograph that I have utilized for twenty years.  Lastly, it is the composition of the samples analyzed that provide the “advanced” dimension of experimental design, not necessarily the particular instrument or experimental system.

“Minor point: please remove double spaces through the whole manuscript.”

I have removed the double spaces between sentences as requested.

Reviewer 3 Report

The manuscript “Ruthenium, not carbon monoxide, inhibits the procoagulant activity of Atheris, Echis and Pseudonajavenoms” provides insight into an understudied phenomenon. The manuscript would be more impactful if the mechanism of action was better known, such as what specific venom component is being inhibited. The ruthenium-mediated procoagulant inhibition could be explored for fractionated venoms, identifying the target enzyme or enzymes. Other major and minor comments to be addressed:

Major:

How does this have applications for snakebite treatment when the effects of ruthenium-based inhibition are absent with albumin? Albumin is a common component of human plasma and therefore, would infer using ruthenium-based envenomation treatments.

Line 70: Although the three chosen venoms all contain serine proteases, metalloproteinases, and phospholipase A2, the abundances of these enzymes are different for these venoms, as well as the number of different isoforms. Is there any relationship between the percentage of these venom components and the inhibition of venom procoagulant activity?

It is also important to note that the proteins responsible for procoagulant activity in snake venoms varies for species. For instance, P. textilis has a unique prothrombin activator complex, which the two viper species in this study do not have. It would be very interesting to fractionate each venom and test individual fractions to identify which venom component is being inhibited by ruthenium.

Line 87-88: Why were different concentrations of venoms used?

Minor:

Figure 1-7: TTG, MRTG and TMRTG should also be defined in the legends. Additionally, it would be beneficial to add the units (dynes/cm2, and etc.) to the y-axis.

Figures 1-3: It would be interesting to see the venoms compared to each other as well. Maybe each variable could show all three venoms in a single graph with different colored bars?

Line 120-137. Needs to not be in italics.

Author Response

“The manuscript “Ruthenium, not carbon monoxide, inhibits the procoagulant activity of Atheris, Echis and Pseudonajavenoms” provides insight into an understudied phenomenon. The manuscript would be more impactful if the mechanism of action was better known, such as what specific venom component is being inhibited. The ruthenium-mediated procoagulant inhibition could be explored for fractionated venoms, identifying the target enzyme or enzymes. Other major and minor comments to be addressed:”

Major:

“How does this have applications for snakebite treatment when the effects of ruthenium-based inhibition are absent with albumin? Albumin is a common component of human plasma and therefore, would infer using ruthenium-based envenomation treatments.”

I appreciate this comment.  First, I plan on the treatment with ruthenium-based antivenom to be regional (bite wound) and not systemic.  Second, the concentration of CORM-2 or other similar compounds can be increased by a factor of ten at the bite would, which is still not toxic.  Thus, the effect of albumin present in tissue can be easily outcompeted and will allow neutralization of the venom.  I have added two sentences to the last paragraph of the discussion section and an additional citation to address this concern.

“Line 70: Although the three chosen venoms all contain serine proteases, metalloproteinases, and phospholipase A2, the abundances of these enzymes are different for these venoms, as well as the number of different isoforms. Is there any relationship between the percentage of these venom components and the inhibition of venom procoagulant activity?”

The reviewer asks an interesting question, but fractionization of the individual venoms and testing each enzyme class with CORM-2 is beyond the scope of the present investigation.  I am simply using venoms that I have published with before as indicated in the citations as a tool to assess if ruthenium species rather than carbon monoxide are responsible for inhibition of the composite prothrombotic activity.  Thus, using this platform, I fairly tested the explicit hypothesis and produced an unambiguous result. 

“It is also important to note that the proteins responsible for procoagulant activity in snake venoms varies for species. For instance, P. textilis has a unique prothrombin activator complex, which the two viper species in this study do not have. It would be very interesting to fractionate each venom and test individual fractions to identify which venom component is being inhibited by ruthenium.”

While I agree with the reviewer that these are interesting matters, they are beyond the scope of the present work.  As a practical matter, I have a laboratory that is coagulation-based and have no capacity to perform the fractionations of the venoms and purify/verify all of the individual enzymes and compounds.  Further, my laboratory is under lock-down secondary to the pandemic and will likely not be operational for at least two months.  Lastly, individual enzymes from these venoms are not commercially available, and I lack the capacity to assess such enzymes for purity.  However, within the confines of the presented experimental conditions, I addressed the hypothesis.  Investigations proposed by the reviewer are likely to be pursued in the years to come by toxinologists that can access purified enzymes from such venoms.

“Line 87-88: Why were different concentrations of venoms used?”

As stated in my manuscript in the first sentence of Results, the concentrations used were the same as that published with them in the past.  These were archived, reconstituted venoms.  The thrombelastographic model is performance-based, and the following sentence has now been placed in Results to provide clarity:

‘Venom concentrations were originally chosen based on a performance basis wherein the activation of coagulation by the venom statistically exceeded the activation observed by contact activation with thrombelastographic cup and pin contact with plasma as previously described [8,10].’

Minor:

“Figure 1-7: TTG, MRTG and TMRTG should also be defined in the legends. Additionally, it would be beneficial to add the units (dynes/cm2, and etc.) to the y-axis.”

I appreciate the reviewer’s suggestions.  I explicitly defined the thrombelastographic variables at the beginning of the results section, and to duplicate this text in every legend would be somewhat repetitive – and in previous works I have been asked to abbreviate such text after reviews are completed.  However, I was happy to revise each graphic and now have all y-axis labeled with the requested units to assist the readership.

“Figures 1-3: It would be interesting to see the venoms compared to each other as well. Maybe each variable could show all three venoms in a single graph with different colored bars?”

While I understand what the reviewer is asking, I respectfully decline for the following reasons.  First, twelve columns across three panels with color will become very complicated and likely loose the readerships interest.  Second, I cannot compare statistically between the three venoms as the concentration chosen was based on a qualitative performance measure wherein each venom’s activation of coagulation had to greatly exceed normal contact activation the thrombelastograph cup and pin – but not to the exact extent.  Thus, I am left with comparing the pattern of inactivation with the ruthenium radical species in a qualitative manner.

It should be noted that I did compare pattern-wise the effects of ruthenium chloride between the venoms, but I did separate the thrombelastographic parameters and limited each venom to two conditions with t-tests to present the information in a clear, uncluttered way likely to be easily interpreted by the readership.

“Line 120-137. Needs to not be in italics.”

This has been corrected.

Round 2

Reviewer 2 Report

1. We could discuss for a long time whether thromboelastography is an advanced technique or not, but this is not the appropriate place. You are right, you provide only mechanistic results which I think is too little to publish in IJMS. The author tries to convince that this particular paper is wise and interesting enough to be published in IJMS, because he has published two more papers in this journal. A very shallow hypothesis, non-justified. Why the author did not use any other methods for kinetic characterization of procoagulant venom? The author should at least discuss this issue.

2. Yes, the author is right. My fault. If the author thinks his explanation and background are enough that is fine. I trust the specialist.

3. Again, this is your field of expertise. But I’ve got one more question – the author tries to persuade that these studies in plasma are sufficient. I cannot agree with this. You cannot omit the tests on the blood. How do you plan to administer these antivenoms in vivo? Separate plasma? Clotting occurs in blood, not only in plasma. What about the role of albumin? Will the author exclude it from blood? Or are you planning to use these antivenoms only locally? If yes it should be discussed.

4. Can the authors include thromboelastographic curves? It would be advisable to see them. Does the author check how AT works in the case of examined venoms? How about the studies in PPP plasma with calcium ions and with low concentrations of thrombin to check the Generation of endogenous thrombin and subsequent clot formation?

5. The author checked RuCl3 at 100 uM, how about the general toxicity of ruthenium on humans? This should be discussed. Please provide also the chemical characterization of CORM-2 and RuCl3 – what was the purity of these agents? Was it validated using HPLC? What about the effects of COMRM-3 and RuCl3 on the fibrinolysis? Did the author perform any studies?

6. Lines 190, 192, 198, 202, 205, 216, 223, 228, 252, 257, 271 – double spaces are still there.

Author Response

All modifications of text have been highlighted in yellow as in the previous revision. I now include a few more references to support my contentions. I now address each comment of the reviewer in detail.

  1. “We could discuss for a long time whether thromboelastography is an advanced technique or not, but this is not the appropriate place. You are right, you provide only mechanistic results which I think is too little to publish in IJMS.”

I must respectfully disagree with the reviewer. My express purposes for conducting this investigation are stated in the Introduction, and as this journal is molecular-based and mechanism-driven, the data provided is quite adequate to be published. The reviewer alludes to the use of more complex systems that simulate either the microcirculation or could involve whole animal models in the previous comments in the first review as well as this one. Not only is this unnecessary, but increasing the complexity of the system can and would abrogate the unambiguous results and the mechanisms revealed with in my investigation.

“The author tries to convince that this particular paper is wise and interesting enough to be published in IJMS, because he has published two more papers in this journal. A very shallow hypothesis, non-justified.”

I again must respectfully disagree with the reviewer. When I noted that previous works of mine have been published in this journal, it was to indicate that the use of my methods and experimental designs have been accepted here as well as at several other journals over the years. My hypothesis was not that if an article is accepted once by a journal, they must accept all other manuscripts as well. As all authors know, this is indeed a null hypothesis.

In the reviewer’s previous assessment of my manuscript it was noted that the reason or hypothesis for the work was not present. I responded that it was contained in the Introduction. It still is, and I have added to it to more explicitly justify my important (not shallow or non-justified) hypothesis in lines 61-65 and 74-76.

“Why the author did not use any other methods for kinetic characterization of procoagulant venom? The author should at least discuss this issue.”

As I noted in my original response to the reviewer, I have used thrombelastography modified in multiple ways to test hypotheses for 20 years. There is no other technique that can assess the interplay of thrombin with fibrinogen and FXIII and determine how the coagulation system is changing. Thrombinoscopes cannot do this, aPTT and PT cannot do this, fibrinogen concentrations and thrombin times cannot do this, platelet aggregation cannot do this, etc. Venoms affect fibrinogen, prothrombin, phospholipids and other components of coagulation that can easily be assess with thrombelastography as I have demonstrated over the past few years. I now include a rather extensive paragraph to clarify why thrombelastography was used in this and previous works as displayed in Discussion in lines 235-266.

  1. “Yes, the author is right. My fault. If the author thinks his explanation and background are enough that is fine. I trust the specialist.”

Thank you.

  1. “Again, this is your field of expertise. But I’ve got one more question – the author tries to persuade that these studies in plasma are sufficient. I cannot agree with this. You cannot omit the tests on the blood.”

As noted, I now provide a very detailed paragraph addressing this issue in lines 235-266. Specifically, I wanted a system with minimal confounding thrombin-generating forces as the venoms are thrombin-generating. The system is optimized to assess the venom’s procoagulant activity, and the addition of kaolin or tissue factor would provide no meaningful information and would only confound the experimental design.

“How do you plan to administer these antivenoms in vivo? Separate plasma? Clotting occurs in blood, not only in plasma.”

The answers to these questions are contained in lines 269-274 where they were originally placed in response to reviewer #3 after the first round of reviews. The antivenom will be administered at the wound site, with the venom’s procoagulant properties neutralized. I never proposed to inject CORM-2 systemically in this or any other manuscript to directly inhibit venom hemotoxic activities.

“What about the role of albumin? Will the author exclude it from blood? Or are you planning to use these antivenoms only locally? If yes it should be discussed.”

I believe that I have already answered this question in lines 269-274.

  1. “Can the authors include thromboelastographic curves? It would be advisable to see them.”

Actually, it is neither advisable or accurate to include this data. As addressed in the new paragraph in Discussion, lines 235-266, the data is nonparametric and at best descriptive. I include my previous work addressing this matter as citation [33].

“Does the author check how AT works in the case of examined venoms?”

The role of antithrombin is irrelevant in the testing of the explicit hypothesis of this work. Given the biochemistry of these venoms, it could be anticipated that addition of a high concentration of heparin could activate endogenous antithrombin activity to the point that the procoagulant effects of the venom could be abrogated – resulting in a complete loss of the ability of the plasma to coagulate. This result is of no use in the determination of the mechanisms responsible for CORM-2 mediated inhibition of these procoagulant venoms.

“How about the studies in PPP plasma with calcium ions and with low concentrations of thrombin to check the Generation of endogenous thrombin and subsequent clot formation?”

Calcium was added to the samples as indicated in the Methods. The venoms are thrombin-generating, and prothrombin cannot be converted to thrombin without sufficient calcium concentrations. As mentioned several times, assessing the activity of the venom’s activity is best performed in an environment with finite but easily outcompeted thrombin generation. The background thrombin generation is assessed in the control samples without venom addition by measurement of corresponding TMRTG, MRTG and TTG values.

  1. “The author checked RuCl3 at 100 uM, how about the general toxicity of ruthenium on humans? This should be discussed.”

The use of ruthenium chloride was to provide mechanistic insight into the effects of the simplest ruthenium (III) compound I could find on the procoagulant activity of the three venoms tested.  There is no imaginable clinical indication to administer this compound systemically - thus, I suspect that there are no phase 1 trials infusing it into animals or humans.  I did not advocate the administration of this compound as a therapeutic – this reviewer is making this claim.

To allay this concern presented by the reviewer, I have added the following sentence to the Introduction, lines 78-80:

‘Critically, the use of RuCl3 in this investigation was purely to provide mechanistic insight, and as there is no clinical indication to administer it to humans or any other species, I am not advocating it as a therapeutic option.’    

“Please provide also the chemical characterization of CORM-2 and RuCl3 – what was the purity of these agents? Was it validated using HPLC?”

CORM-2 was quality assured with infrared spectrum analysis, with conformity to structure within acceptable limits.  This is public domain material that can be observed at Sigma Aldrich.  Ruthenium chloride was 99.98% pure based on trace metal analysis as per the certificate of analysis that is public domain, obtainable from Sigma Aldrich.

With regard to CORM-2, this is the source and specific reagent that I have used for the last 12 years – its effects on plasma, whole blood, and numerous isolated compounds and enzymes has not been appreciably different over this time.  The need to further analyze these compounds obtained from this internationally respected vendor by HPLC (with what detector – MS, electrochemical or other spectrophotometer?) is beyond the scope of the present work.

“What about the effects of COMRM-3 and RuCl3 on the fibrinolysis? Did the author perform any studies?”

Again, the experimental design was laser-focused on assessing if carbon monoxide or ruthenium has been the agent that inhibits the procoagulant activity of the three venoms tested.  There was no need to add an additional series of experiments using tPA or any other fibrinolytic as they would provide no additional mechanistic insight into my hypothesis.

  1. “Lines 190, 192, 198, 202, 205, 216, 223, 228, 252, 257, 271 – double spaces are still there.”

These have all been corrected.

In closing, I would appeal to the reviewer to carefully consider my comments and hopefully appreciate the thought and detail that went into the experimental design, execution, and interpretation of the data contained in this manuscript. Determining if CORM-2 is mediating its effects by Ru-based radicals rather than carbon monoxide release has far reaching implications – and has the potential to cause many investigators to reconsider their mechanistic interpretations.

Reviewer 3 Report

All my concerns have been addressed. Additional clarifications that have been added have improved the manuscript.

Author Response

Thank you for your kind comments that improved my manuscript.

Round 3

Reviewer 2 Report

I have no more questions, although not all issues were addressed.